# Variation of Saponins in *Sanguisorba officinalis* L. before and after Processing (*Paozhi*) and Its Effects on Colon Cancer Cells In Vitro

**DOI:** 10.3390/molecules27249046

**Published:** 2022-12-19

**Authors:** Zhengyang Wang, Chunjuan Yang, Lihong Wu, Jiahui Sun, Zhenyue Wang, Zhibin Wang

**Affiliations:** 1Key Laboratory of Basic and Application Research of Beiyao, Ministry of Education, Heilongjiang University of Chinese Medicine, Harbin 150040, China; 2Department of Pharmaceutical Analysis and Analytical Chemistry, College of Pharmacy, Harbin Medical University, Harbin 150086, China

**Keywords:** activity study, potential ingredients, traditional Chinese medicine, qualitative analysis, content determination, UHPLC-MS/MS

## Abstract

The incidence of colon cancer is increasing year over year, seriously affecting human health and quality of life in recent years. However, traditional Chinese medicine (TCM) has been utilized for the treatment of colon cancer. *S. officinalis* Saponins (*S-Saponins*), the potential compound of TCM, displays multiple biological activities in colon cancer treatment. In our study, ultra-high-performance liquid chromatography–tandem mass spectrometry (UHPLC-MS/MS) combined with multivariate statistical analysis were performed to analyze and identify raw and processed saponins. Then, MTT and cell migration assays were used to preliminarily explore the effects of saponins in vitro on colon cancer cells. The results showed that 29 differential saponins compounds under *Paozhi* were identified by UHPLC-MS/MS. Moreover, in vitro validation showed that *Sprocessed* better inhibited the proliferation and migration of colon cancer cells than *Sraw*. This study provides a basis for the determination of the chemical fundamentals of the efficacy changes during *Paozhi* through inferring the changes in saponin components and its possible transformation mechanisms before and after processing *S. officinalis*. Meanwhile, it also provides new insights into potential bioactive ingredients for the treatment of colon cancer.

## 1. Introduction

In recent years, colon cancer has become a common malignant tumor of the digestive tract, which seriously threatens human health [1]. Early diagnosis of colon cancer can effectively reduce the incidence and mortality of colon cancer, which will mean that more patients with colon cancer will have the opportunity to have a better chance of long-term survival due to early detection, early diagnosis, and early treatment. In addition, it has been reported that the five-year survival rate of patients with colon cancer is more than 90% if it is prevented and treated in an early stage [2]. Currently, there are several kinds of early diagnosis methods for colon cancer, including the fecal immunochemical test (FIT), colonoscopy, a guaiac-based fecal occult blood test, multitargeted stool DNA test (FIT-DNA), flexible sigmoidoscopy, CT colonography, etc. [3]. In addition, Annamaria pallag et al. used immunohistochemical and histoenzymatic techniques to monitor colon cancer cells for achieving the purpose of diagnosis [4].

At present, the treatment of colon cancer mainly focuses on surgery, radiation therapy, and chemotherapy. However, the serious side effects of chemotherapy drugs and other disadvantages limit its clinical application [5]. Traditional Chinese medicine (TCM) has shown anti-tumor potential through “multi-component, multi-target, and multi-pathway” characteristics compared with chemotherapy drugs, which can improve the disease process through a variety of ways [6]. Furthermore, researchers have been attracted by TCM’s low cost, stable curative effect, and minimal side effects, leading them to explore TCM’s potential and application prospects for treating tumors [7]. Therefore, it is urgent to find natural medicine to meet the treatment needs of the complex pathological mechanism of colon cancer and lower the risk of colon cancer.

Saponins are one of the main bioactive ingredients in natural medicine. In recent years, the pharmacological effects of saponins in the treatment of colon cancer have received extensive attention [8,9,10]. It has been found that ginsenosides, paris saponins, multiple saponins, etc., can effectively inhibit the proliferation of colon cancer cells, which indicates that new active compounds against colon cancer can be extracted from natural medicine rich in saponins [11,12,13]. *Sanguisorba officinalis* L. (the dried root of Sanguisorba, Rosaceae, *S. officinalis*) as TCM contains abundant saponins compounds and is widely distributed in Europe and Asia [14,15]. In addition, multiple studies showed that triterpenoid saponins of *S. officinalis* (*S-Saponins*), the potential compound of TCM, possesses multiple biological activities and application prospects [16,17]. Therefore, it is very necessary to clarify the types of triterpenes saponins and enrich the chemical composition and activities studied in *S. officinalis*.

In recent years, with the development of technology and changes in processing technology, the clinical application of *Sanguisorba officinalis* L. saponins-processed (*Sprocessed*) products has also been deepened [18,19]. The ancient processing methods of *S. officinalis* include roasting, frying, simmering, and using vinegar, wine, charcoal, etc. [20]. Among them, raw products and charcoal products have been used to this day [21]. The chemical composition and trace elements contained in *S. officinalis* change after processing. Traditionally, it is believed that *S. officinalis* enhances its corresponding efficacy after processing [22]. TCM has very complex compounds and the chemical compositions may change after processing; therefore, it is important to analyze the changed saponins components before and after processing to clarify their pharmacodynamic material basis. At present, ultra-high-performance liquid chromatography–tandem mass spectrometry (UHPLC-MS/MS) technology has been used to study the active ingredients of complex systems of natural medicines, given its high sensitivity and resolution, good selectivity, and short analysis time [23,24]. However, the studies on the composition changes of *S. officinalis* before and after processing mainly focus on a certain component [25,26], and there are few reports of studying multiple component changes simultaneously. Obviously, the pharmacological activity and intrinsic quality of natural medicine cannot be fully described by considering only one compound. Therefore, it is necessary to monitor as many bioactive components of *S. officinalis* under *Paozhi* as possible to ensure its quality and efficacy. We can more fully reflect the differences between pharmacodynamic substances before and after processing by accurately controlling the changes in saponin components and screening different compounds.

To the best of our knowledge, no study has comprehensively analyzed the different compounds in raw and processed *S. officinalis*. In our study, for the first time, the differential components within differently processed *S. officinalis* saponins products were analyzed comprehensively, and the possible reasons for the differences in efficacy were considered. Meanwhile, this study also contains some innovative content in the application 0of mass spectrometry. The instrumentation of UHPLC-MS/MS was used to rapidly detect and identify the components in *S. officinalis* and, at the same time, a multivariate statistical analysis approach was used to discover the changes in chemical composition after processing. Consequently, a convenient and systematic way was established to investigate the changes in different compounds of *Sraw* and *Sprocessed*. Meanwhile, this study could serve as a theoretical basis for intensive mechanistic studies of *S. officinalis* processing and reasonable clinical applications. Moreover, we preliminarily evaluated the effects of *Sraw* (*Sanguisorba officinalis* L. saponins raw) and *Sprocessed* (*Sanguisorba officinalis* L. saponins processed) on the growth of colon cancer cell lines. From the *Paozhi* perspective, we tried to reveal the changed compounds and possible transformation pathways in *S. officinalis*, which may be the core link to help explain its use for colon cancer prevention. Furthermore, it also might promote the development of effective disease-modifying TCM extracts.

## 2. Results

### 2.1. Optimization of Chromatographic and Mass Spectrometric Conditions

To obtain good separation of various compounds in raw and processed *S. officinalis*, the chromatographic conditions were optimized by using different compositions of mobile phase and adjusting the gradient elution. After investigating a range of mobile phase systems, including methanol–water, acetonitrile–water, and methanol–0.1% aqueous formic acid, acetonitrile–0.1% aqueous formic acid was chosen because it provided symmetric peak shapes. Then, the gradient elution was examined to ensure peak resolution. Because of the complexity of TCM ingredients, to achieve as much mass spectrometry (MS) information as possible, the compounds were investigated under positive and negative-ion modes. Triterpenoid saponins of *S. officinalis* had a strong response in positive and negative-ion modes, which provided the low background noise and high sensitivity. Collision energy sets of 5/−5 eV, 15/−15 eV, 25/−25 eV, and 35/−35 eV were used to ensure that the precursor ions could be properly dissociated. Therefore, enough product ions to enable subsequent chemical structure analysis could be obtained. The ion spray voltage, turbo spray temperature, declustering potential, and the nebulizer gas, heater gas, and curtain gas pressure were optimized to obtain better ionization efficiency and more chemical structure information.

### 2.2. Identification of Compounds in Sraw and Sprocessed

In this study, scan mode and production mode were used for detection under the above optimized chromatographic and MS conditions. The molecular weights of the S-saponins were determined by analyzing the mass spectra of the chromatographic peaks obtained in the full-scan chromatograms under positive and negative-ion modes. Additionally, the MS and MS/MS spectra of each constituent were analyzed through database searches, comprehensive mass spectrometry data and comparison of literature to infer the chemical structures of some of the compounds. [27,28,29]. The structures of some kinds of compounds can be further confirmed by comparing with the retention time of the reference substance and the mass spectrometry data. As a result, it was found that there are 27 compounds in the *Sraw* and 48 compounds in the *Sprocessed* under the negative-ion detection mode. Among of them, 15 common saponins compounds were identified by matching the empirical molecular formula with that of the published compounds or available reference compounds. The MS total ion current (TIC) chromatogram of both *Sraw* and *Sprocessed* under negative-ion mode is shown in Figure 1. Simultaneously, the *Sraw* (17 compounds) and *Sprocessed* (32 compounds) were detected under the positive-ion detection mode. Among of them, eight common saponins compounds were identified. Additionally, the TIC of the positive-ion mode is shown in Figure 2. According to retention time, the common 23 compounds were arranged from front to back. Additionally, the details are summarized in Table 1 (15 compounds under the negative-ion mode) and Table 2 (8 compounds under the positive-ion mode). Moreover, 12 compounds disappeared after being processed in the positive- and negative-ion modes. Additionally, among of them, three compounds were identified, including one kind of compounds under positive-ion mode and two under negative-ion mode. The information of mass spectrometry fragmentation is shown in Table 3, and the specific structure and mass spectrum of the three disappeared compounds are shown in Figure 3. Simutaneously, 48 new compounds were added after processing. Additionally, among of them, five compounds in the negative-ion mode were identified. Then, in the positive-ion mode, four compounds were also identified. The information of mass spectrometry fragmentation is shown in Table 4, and the specific structures and mass spectra of nine newly added compounds are shown in Figure 4. The chemical composition of *Sprocessed* has a certain degree of change, including the changes of newly added and disappeared compounds compared with Sraw.

### 2.3. Quantitative Determination in Sraw and Sprocessed

In order to screen differential compounds, in our study, firstly, we have compared the content of saponins ingredients in *Sraw* and *Sprocessed* through the peak area change index (processed peak area/raw peak area). When the change index was a positive value, it meant that the content of compounds increased after being processed. On the contrary, the compounds content decreased. The closer it was to the abscissa, the less the content changed. According to the histogram in Figure 5 and the cluster heat map in Figure 6, there were certain differences in the content of each compound before and after being processed. The contents of eight kinds of di-substituted and mono-substituted saponins, such as N6, N7, N10, N11, N16, N18, P1, and P8 in *S. officinalis* decreased after being processed, while the aglycone generally increased. The component analysis showed that most of the saponins in “charcoal” increased; however, there were also some saponins with decreased content. The reason may be that the transformation of different degrees occurred between the compounds during processing. It also may produce the key component for reducing side effects and increasing efficiency. In this study, we speculate the possible transformation pathways under processing, as shown in Figure 7. Interestingly, by comparing the structures of the changed pentacyclic triterpene saponins, we found that “charcoal” provided heating conditions, which resulted in three reactions of pentacyclic triterpene saponins: sugar chain breakage (discontain oxygen—the fracture location was at the red dotted line), glycosidic bond breakage (contain oxygen—the fracture location was at the green dotted line), and glycosyl ring-opening cleavage. Based on the above principles, during the processing, the disubstituted saponins may first be converted into the monosubstituted components and then into the sapogenin components. These chemical reactions change the structure–activity groups of pentacyclic triterpene saponins. Then, it may cause the changes in content and pharmacological effects, with differences in the component content of *Sraw* and *Sprocessed* also affecting the efficacy. Therefore, in the follow-up experiment, we used multivariate statistical analysis to search for marker compounds that might have different efficacy.

### 2.4. Multivariate Statistical Analysis

We preprocessed the liquid and mass data, such as the retention time, mass-to-charge ratios, and peak area before and after the processing of *S. officinalis* to group for marker compounds. Then, we used SIMCA 14.0 software for principal component analysis (PCA), as shown in Figure 8A. Two-dimensional PCA score plots in positive- and negative-ion modes exhibited a tendency to separate the raw and processed *S-saponins*. Additionally, two-dimensional loading plots in positive- and negative-ion modes showed components that made markable contributions toward discriminating raw and processed *S. officinalis*. The discreteness between the sample points in each group was small, indicating good homogeneity within the group. The raw and processed *S-saponins* gathered in the same area, which further indicated that there were some differences in the chemical constituents of *S. officinalis* before and after processing. In order to better observe the differences between groups before and after processing, we supervised orthogonal partial least squares-discriminant analysis (OPLS-DA), which was further carried out to obtain the corresponding model on the basis of PCA. The OPLS-DA score is shown in Figure 8B. Among of them, R1–R6 (Raw1–Raw6) were located on the right side of the score map, and P1–P6 (Processed1–Processed6) were located left side, indicating that the established model can be effectively used for the analysis of chemical components before and after processing. In order to further study the contribution of each compound during processing, the OPLS-DA variable importance in projection (VIP) value was used to screen the differential markers. A higher VIP value indicated that the corresponding compound contributes more to the mass difference. Therefore, the differential compounds undergoing processing can be effectively screened by screening the variables with VIP > 1. The results showed that there were 17 different compounds before and after processing, which were N12, P9, P10, P4, P7, N2, N18, N14, N6, N15, P8, N13, N19, N11, N20, N5, and P2. These 17 compounds may be the iconic compounds that cause the changes. In addition, the discovery of these iconic compounds is conducive to the study of TCM monomers or the active ingredients of the extracts. However, due to the incomplete literature reports on the compounds of *S. officinalis*, many other substances corresponding to chromatographic peaks have not been identified, and further in-depth research is still required.

### 2.5. Effects of Sraw and Sprocessed on the Growth of Colon Cancer Cells In Vitro

To determine the anti-colon cancer effect of *Sraw* and *Sprocessed*, two different types of human colon cancer cell lines (HCT116, RKO) were treated with *Sraw* and *Sprocessed* (20, 40, 80, 160, 320, 640, and 1280 μg/mL). The MTT assay showed that after treatment for 24 and 48 h, *Sraw* and *Sprocessed* had a significant inhibitory effects on human colon cells, which gradually increased with increasing drug concentrations. However, *Sprocessed* was better inhibited on HCT116 and RKO cells, as shown in Figure 9A,B. The IC_50_ values of the HCT116 and RKO were 609.0, 544.8 μg/mL, and 458.3, 451.3 μg/mL at 24 and 48 h after treatment with Sraw, respectively. IC_50_ values of the HCT 116 and RKO were 562.5, 512.1 μg/mL and 342.4, 416.5 μg/mL at 24 and 48 h after treatment with *Sprocessed*, respectively. Based on these results, we selected *Sraw* and *Sprocessed* concentrations of 80, 160, and 320 μg/mL for the subsequent experiments.

### 2.6. Sraw and Sprocessed Inhibited the Migration of HCT116 and RKO Cells

To explore the underlying the inhibited effects of *Sraw* and *Sprocessed* on HCT116 cells and RKO cells, we conducted cell migration assays. In order to observe the migration ability of the cells, cell scratch experiments were used to measure the migration distance of the cells. To avoid the cells of the control group becoming connected in clusters, we selected 24 h as the experiment duration. The results showed that *Sraw* and *Sprocessed* inhibited the movement of HCT116 and RKO cells in a dose-dependent manner, and the scratching width of the cells in the treatment group was significantly greater than that of the control group after 24 h. Compared with the control group, HCT116 cells treated with 80, 160, and 320 μg/mL concentrations of *Sraw* and *Sprocessed* showed mobilities of 52.83%, 44.52%, and 22.89% (*Sraw*) and 48.92%, 42.60%, and 21.04% (*Sprocessed*) (Figure 10). RKO cells treated with 80, 160, and 320 μg/mL concentrations of *Sraw* and *Sprocessed* showed mobilities of 45.55%, 30.14%, and 25.93% (*Sraw*) and 43.01%, 29.97%, and 22.90% (*Sprocessed*), which represented a significant effect (*p* < 0.001) (Figure 11). Based on these results, *Sprocessed* better inhibited the migration of HCT 116 and RKO cells compared with *Sraw*.

## 3. Discussion

In this study, 29 different saponins were identified, including N12, P9, P10, P4, P7, N2, N18, N14, N6, N15, P8, N13, N19, N11, N20, N5, P2 (content difference compounds), N1, N3, P3 (disappeared compounds), N11, N12, N13, N15, N26, P4, P7, P19, and P20 (new compounds) compared with the previous studies about the *Paozhi* of saponins. In addition, the in vitro cell experiment also verified that *Sprocessed* had better anti-colon cancer cell efficacy than *Sraw*. It shows that *Paozhi* would bring new changes to the composition and anti-tumor efficacy of *S-Saponins*. These new findings were mainly related to the following: First, the high sensitivity and dynamic background deduction function, as well as the accompanying target-compound screening function of UHPLC-ESI-MS, enabled the detection of extremely low concentrations, and new and disappeared compounds, even though they interfered with the complex mechanism background. The second point is that due to the influence of *Paozhi* technology, the changes to the saponin components increased, reflecting the overall difference in the composition of *S-Saponins* before and after processing.

In addition, the most direct purpose of TCM processing is to reduce the side effects of the drug and enable people to use the drug safely; in other words, to improve the efficacy of compounds as much as possible through a reduction or increase in the quantities of certain ingredients [30,31,32]. However, since the toxic compounds in many Chinese herbal medicines also have certain medicinal effects, the purpose of our study is to control the compounds within a safe and reasonable range so that they can be effective at the same time [33,34]. In our study, the *Paozhi* technology of “charcoal” was used, which was mainly to destroy the structure of medicinal compounds by increasing the temperature so that some compounds were converted into other fewer-side-effects-inducing substances to achieve the purpose of improving efficacy. The changes in chemical compounds during processing also provided the chemical basis for subsequent pharmacodynamic studies in vitro.

Meanwhile, we processed the *S. officinalis* according to the method in the Chinese pharmacopoeia and applied the established content determination method to determine *Sraw* and *Sprocessed* and compare the content changes [35]. From the perspective of the overall compound changes, 17 different compounds were screened through UHPLC-MS/MS combined with multivariate statistical analysis. Among of them, the contents of di-substituted and mono-substituted saponins in *S. officinalis* were decreased after being processed, while the aglycone was generally increased. The changes in the content of these compounds before and after being processed may provide some ideas for the study of the discovery of TCM monomers or the active ingredients of extracts. However, most of the differential compounds obtained by screening lack relevant efficacy research reports, which may also cause the internal reasons for the differences in pharmaceutical effects before and after processing. Therefore, in our study, we combined cells in vitro to provide novel insights for *S-Saponins* before and after *Paozhi*.

It has been reported that saponins have obvious inhibitory effects on colon cancer cells [36,37]. Both MTT and migration assays confirmed that *Sprocessed* had better therapeutic effects during in vitro cell experiments. The reason may be that there is a certain degree of difference in the chemical composition of *Sraw* and *Sprocessed.* Meanwhile, in our study, we have already identified different compounds before and after processing, including 3 disappeared compounds, 9 new compounds, and 17 kinds of iconic compounds through UHPLC-MS/MS technology, which may finally cause the difference efficacy on cells in vitro.

In this study, an accurate, reliable and simple method was established for the analysis of the saponins of *S. officinalis* before and after processing to illuminate the composition differences and content changes in saponins. Moreover, the differences between *Sraw* and *Sprocessed* against colon cancer were verified by simple in vitro cell experiments, which provided a certain guiding significance for the discovery of different compounds of *S. officinalis*. Meanwhile, the results of our study would be a helpful reference for further explaining the mechanisms underlying the effects of the processing process on *S. officinalis*, helping us to find the potential bioactive components for treating colon cancer. In addition, our study suggested that the UHPLC-MS/MS method was useful in studies of active compounds in TCM extracts. However, its anti-tumor mechanism still needs to be further studied. In the next step, we will carry out the research of these different compounds to find out the in vivo processes and mechanisms for treating colon diseases.

## 4. Materials and Methods

### 4.1. Materials and Reagents

UHPLC-ESI-MS/MS was performed on an Agilent 1290 ultra-high-performance liquid chromatography (UHPLC) system and an Agilent 6430 QQQ-MS mass spectrometer with an electrospray ionization (ESI) source interface (Agilent Technologies, Santa Clara, CA, USA). Additionally, an ACQUITY UPLC BEH Shield RP18 column (50 × 2.1 mm, 1.7 μm) was selected. The standard substances of ziyuglycoside II, 3*β*,19*α*-dihydroxyurs-12-en-28-oic-acid-28-*β*-D-glucopyranosyl ester, 3-β-O-α-L-arabinosylurs-12,18(19)-dien-28-acid-β-D-glucose, ziyuglycoside I, Euscaphic acid, and 1β-hydroxyrosic acid were defined in our laboratory (identified by NMR and MS). The purities of all references were determined to be more than 98%. Formic acid (purity ≥ 99%) of HPLC-grade was purchased from Kermel (Tianjin, China). Methanol and acetonitrile of HPLC-grade were purchased from Dikma Tevhnologies Inc. (Beijing, China). Ultra-pure water used throughout the experiment was prepared from a MilliQ water purification system (Millipore, Molsheim, France). *S. officinalis* was purchased from the Anguo Traditional Chinese Medicine Market of Baoding (Hebei, China). Human colon cancer cell line HCT116 and RKO were purchased from Pricella (Wuhan, China). Roswell Park Memorial Institute (RPMI) 1640 culture medium was obtained from Gibco (Grand land, NY, USA). DMSO was obtained from Sigma-Aldrich (St. Louis, MO, USA). 3-(4,5-dimethylthiazol-2-yl)-2,5-diphenyltetrazolium bromide (MTT) was obtained from Sigma (Burlington, NJ, USA).

### 4.2. Preparation of Sanguisorba officinalis L. before and after Samples

According to the processing method mentioned in the 2020 edition of the “Chinese Pharmacopoeia”, it was prepared into *S. officinalis* charcoal, as follows: Put the clean *S. officinalis* in a hot pot, and fry it with a strong fire until the surface is burnt black. When its interior is brown, spray a small amount of water on it, extinguish the sparks, take it out, and dry it to obtain *S. officinalis* charcoal.

After crushing the dried root of *S. officinalis* (100 g), it was extracted by hot reflux with 0.8 L of a 70% ethanol–water solution for 60 min at 80 °C, thrice in total. All the decoctions were combined, filtered, and then evaporated into steam. The residue was added to an appropriate amount of distilled water for complete suspension, extracted with n-butanol solution for three times, and all the supernatant was extracted. We combined the filtrates in a 10 mL volumetric flask and added to the volume with methanol. All solutions were filtered through filter membrane with pore size of 0.22 µm before use.

The preparation method for the processed *S. officinalis* sample was the same as for the raw *S. officinalis* sample.

### 4.3. Preparation of Standard Solutions

For the qualitative identification of the main compounds of *S. officinalis.*, the standard stock solutions of 6 reference standards (ziyuglycoside II, 3*β*,19*α*-dihydroxyurs-12-en-28-oic-acid-28-*β*-D-glucopyranosylester,3-β-O-α-L-arabinosylurs-12,18(19)-dien-28-acid-β-D-glucose, ziyuglycoside I, Euscaphic acid, and 1β-hydroxyrosic acid) were prepared by dissolving them in methanol, respectively. Additionally, then, the appropriate amount of each standard stock solution was taken, mixed, and finally diluted to an appropriate concentration for further analysis.

### 4.4. Liquid Chromatographic Conditions

The chromatographic separation equipment used was the Agilent 1290 ultra-high-performance liquid chromatography (UHPLC) system. The chromatographic column was an ACQUITY UPLC BEH Shield RP18 column (50 × 2.1 mm, 1.7 μm). The mobile phase of the eluent was 0.1% formic acid in water (A) and acetonitrile (B), and the flow rate was 0.3 mL/min. The column temperature was 30 °C. To obtain a better liquid–phase separation effect, different solvents and gradient profiles of the mobile phase were investigated. The optimum gradient elution program was set as follows: 0~5 min, 5–15% B; 5~15 min, 15–30% B; 15~20 min, 30% B; 20~25 min, 30–40% B; 25~30 min, 40% B; 30~35 min, 40–55% B; 35~45 min, 55% B; 45~50 min, 55–80% B; 50~55 min, 80% B. 5 µL of the sample solution and reference substance solution was respectively injected into the UHPLC-MS/MS system for analysis.

### 4.5. MS Spectrometry Conditions

The mass spectrometer was operated in positive- and negative-ion mode during each detection procedure. The analytical conditions were as follows: The capillary voltage was set at 4500 V in positive-ion mode and the capillary voltage was set at −3500 V in negative-ion mode. The source temperature was kept at 100 °C and the desolvation temperature was kept at 350 °C. The mass range recorded *m*/*z* 100–1500. Nitrogen was selected as drying gas at a flow rate of 11 L/min, high-purity Nitrogen was used as the nebulizing gas.

### 4.6. Cell Culture

HCT116 and RKO cells were maintained in RPMI 1640 medium supplemented with 10% fetal bovine serum and 1% penicillin. The cells were then cultured at 37 °C in a humidified atmosphere containing 5% CO_2_ [38].

### 4.7. Cell Viability Assay

Cell viability was determined using the MTT assay. HCT116 cells and RKO cells (2 × 10^4^) were seeded in 96-well plates for 24 h and treated with different compounds at various concentrations of *Sraw* and *Sprocessed* (20, 40, 80, 160, 320, 640, and 1280 μg/mL) for 24 h and 48 h, respectively. The medium was removed and washed with PBS once before incubating the cells with MTT solution at the final concentration of 20 μL for 4 h. DMSO-dissolved formazan was read at the wavelengths of 570 nm. The cell viability was calculated using the following equation:

Cell viability (%) = (drug treatment group-blank control group)**/**(normal control group-blank control group) × 100 [39].

### 4.8. Cell Migration Analysis

HCT116 cells and RKO cells were cultured in six-well plates at 3000 cells per well, respectively. A total of 200 µL was pipetted perpendicular to the orifice plate from top to bottom to make scratches. Then, the medium was discarded and *Sraw* and *Sprocessed* were added at different concentrations. After 24 h, cells were washed twice with PBS. Finally, the numbers of migrating cells in each group were calculated using the ImageJ Pro software [40].

## 5. Conclusions

This study demonstrated that qualitative and quantitative analysis methods were successfully developed to screen the content changes and iconic compounds of *S. officinalis* before and after *Paozhi* through UHPLC-ESI/MS combined with multivariate statistical analysis technology. Then, in vitro validation showed that *Sprocessed* was better at effectively inhibiting the proliferation and migration of colon cancer cells than *Sraw.* Additionally, the changes in *Saponins* composition and content under *paozhi* may be the core reason for the differences in efficacy. This study systematically studies the saponin compounds of *S. officinalis* before and after processing, and provides an effective analytical strategy for the rapid screening and identification of saponins. In addition, the results of this study provided a theoretical basis for quality control and pharmacological application research, and the possible transformation mechanisms during processed *S. officinalis* were also preliminarily discussed. Meanwhile, this study also laid a chemical basis for the discovery of bioactive components and the clinical application of *S-saponins* in the treatment of colon cancer.

## Figures and Tables

**Figure 1 molecules-27-09046-f001:**
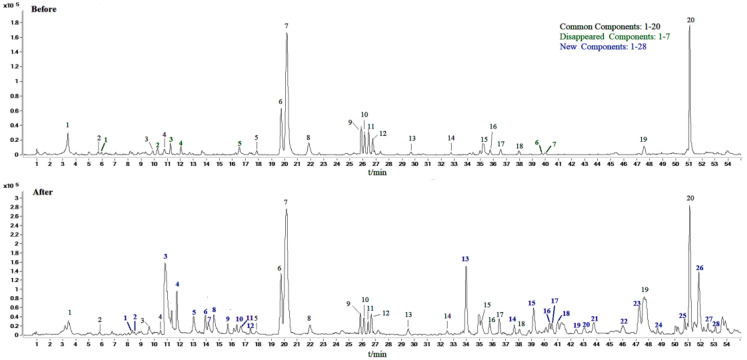
Negative-ion chromatogram of *Sanguisorba officinalis* L. before (*Sraw*) and after (*Sprocessed*) processing.

**Figure 2 molecules-27-09046-f002:**
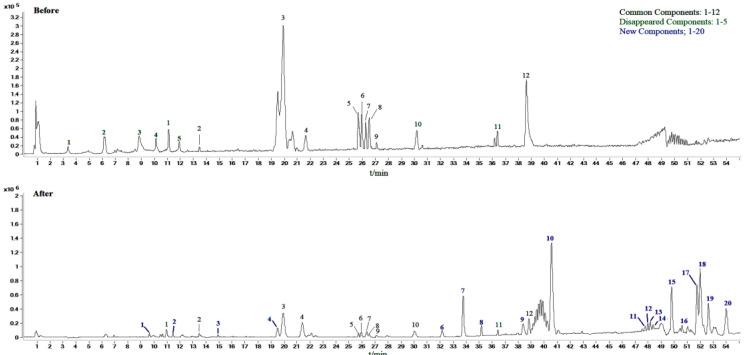
Positive-ion chromatogram of *Sanguisorba officinalis* L. before (*Sraw*) and after (*Sprocessed*) processing.

**Figure 3 molecules-27-09046-f003:**
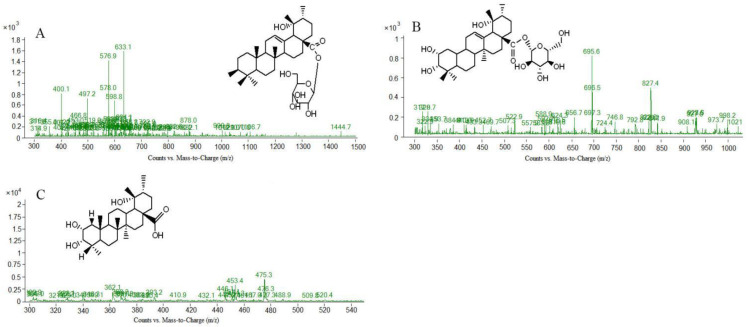
Mass spectrum and structure of disappeared components in *Sanguisorba officinalis* L. after processing. (**A**): 3β, l9α-dihydroxyursin-12-en-28-acid-β-D-glucose ester; (**B**): 2α,3α,19α-trihydroxyurs-12-en-28-acid-β-D-glucopyranosyl ester or isomer; (**C**): 2α,3α,19α-trihydroxyurs-12-en-28-oic-acid.

**Figure 4 molecules-27-09046-f004:**
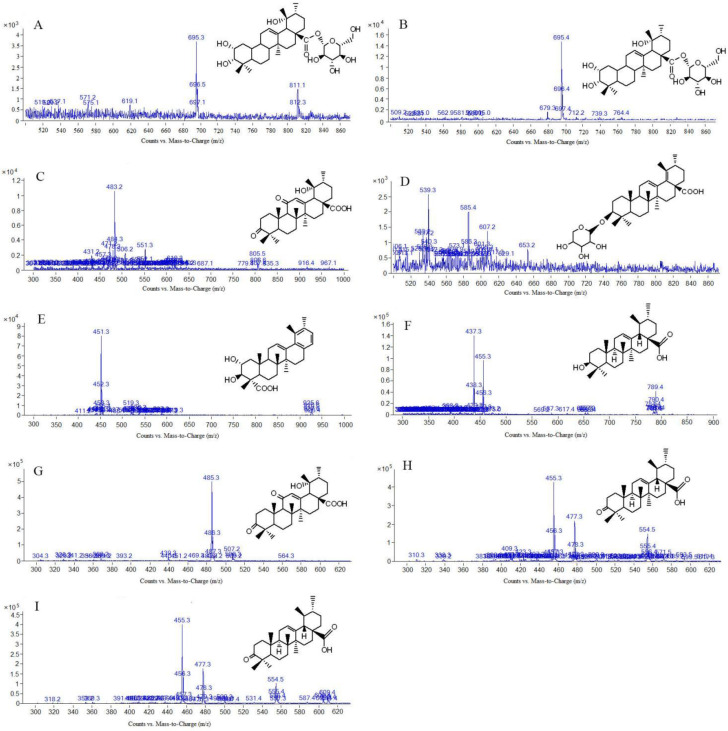
Mass spectrum and structure of new components in *Sanguisorba officinalis* L. after processing. (**A**): 2α,3,19-Trihydroxyurs-12-en-28-acid-β-D-glucopyranosyl ester or isomer; (**B**): 2α,3,19-Trihydroxyurs-12-en-28-acid-β-D-glucopyranosyl ester or isomer; (**C**): 3,11-dioxo-19α-hydroxy-urs-12-en-28-oic acid; (**D**): 3β-O-α-L-arabinopyranosylusr-12,18-dien-28-acid; (**E**): 2α,3β-dihydroxy-28-norurs-12,17,19(20),21-tetraen-23-oic acid; (**F**): ursolic acid; (**G**): 3,11-dioxo-19α-hydroxy-urs-12-en-28-acid; (**H**): 3-oxo-12-en-28-ursolic acid or isomer; (**I**): 3-oxo-12-en-28-ursolic acid or isomer.

**Figure 5 molecules-27-09046-f005:**
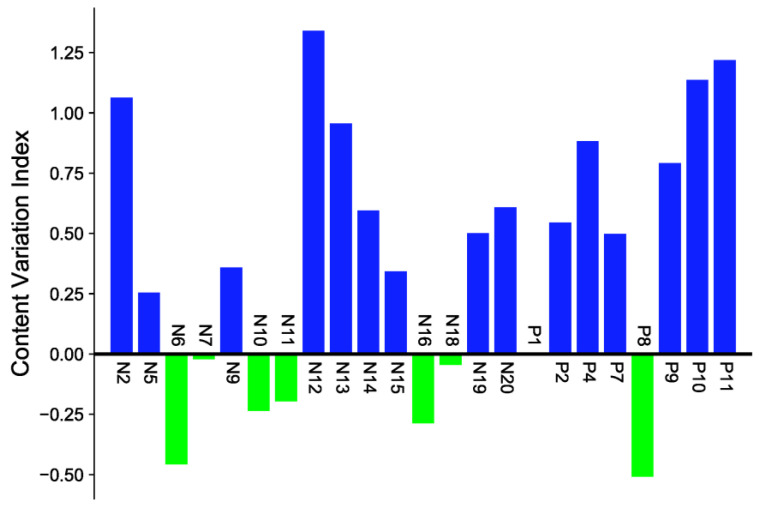
Content change of chemical components in *Sanguisorba officinalis* L. before and after processing.

**Figure 6 molecules-27-09046-f006:**
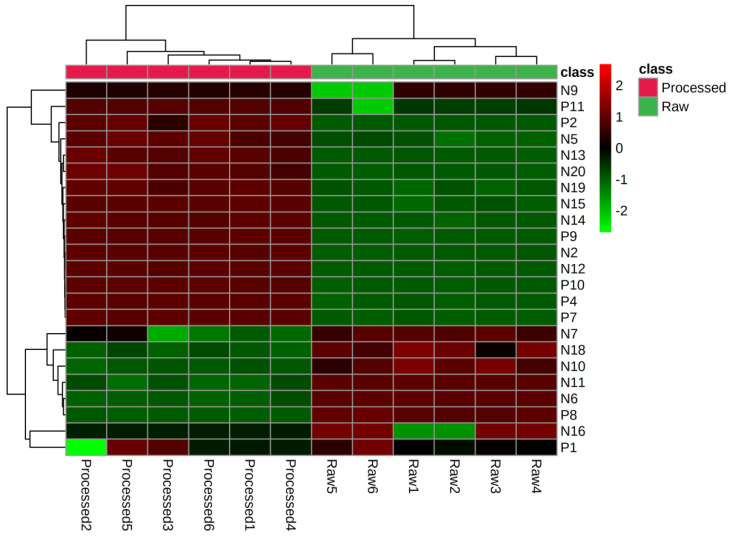
Cluster heat map of *Sanguisorba officinalis* L. before and after processing.

**Figure 7 molecules-27-09046-f007:**
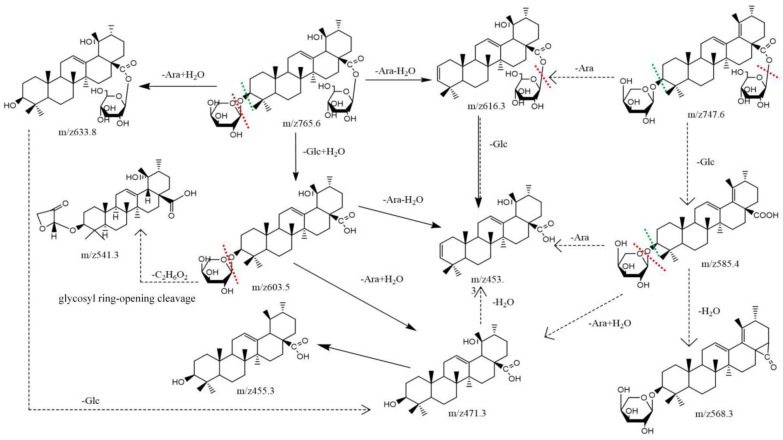
The possible transformation process in *Sanguisorba officinalis* L. under processing.

**Figure 8 molecules-27-09046-f008:**
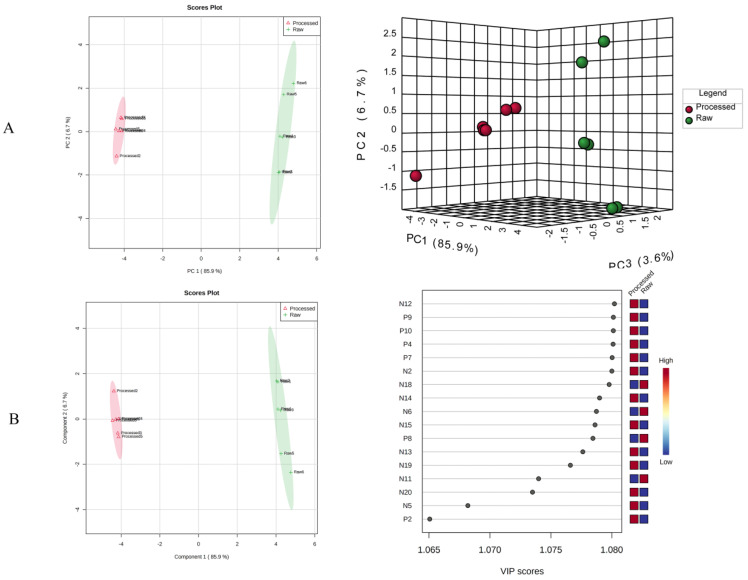
Multivariate statistical analysis score graph. (**A**): PCA scores of raw and processed *Sanguisorba officinalis* L. (2D, 3D); (**B**): The OPLS-DA model and VIP values of differential components in *Sanguisorba officinalis* L. before and after processing.

**Figure 9 molecules-27-09046-f009:**
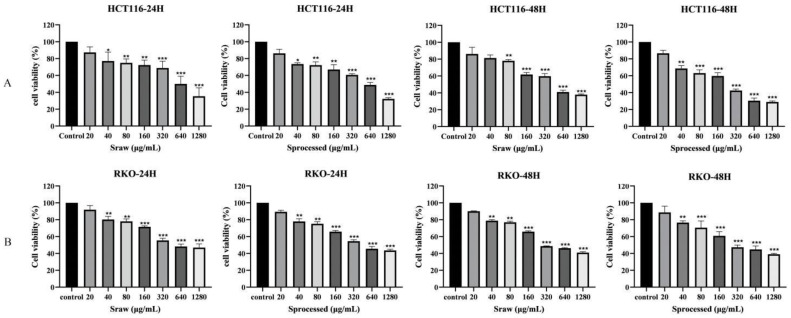
*Sraw* and *Sprocessed* inhibited the proliferation of human colon cancer cells. (**A**): Inhibition of the proliferation of HCT 116 colon cancer cells by different concentrations of *Sraw* and *Sprocessed*; (**B**): Inhibition of the proliferation of RKO colon cancer cells by different concentrations of *Sraw* and *Sprocessed*. Data are expressed as mean ± SEM (*n* = 6); “*” indicates the difference between one group and the control group, respectively. * *p* < 0.05, ** *p* < 0.01, *** *p* < 0.001 vs. 0.0 μg/mL.

**Figure 10 molecules-27-09046-f010:**
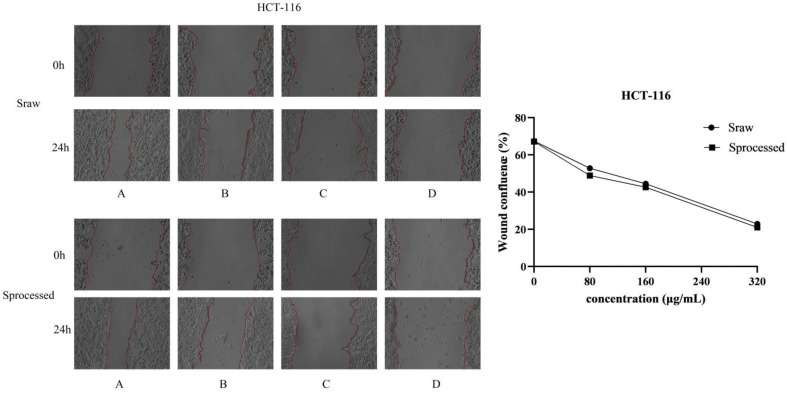
Wound healing assays in HCT116 cells showing inhibition by *Sraw* and *Sprocessed* (scratch analysis). (**A**): 0 μg/mL; (**B**): 80 μg/mL; (**C**): 160 μg/mL; (**D**): 320 μg/mL.

**Figure 11 molecules-27-09046-f011:**
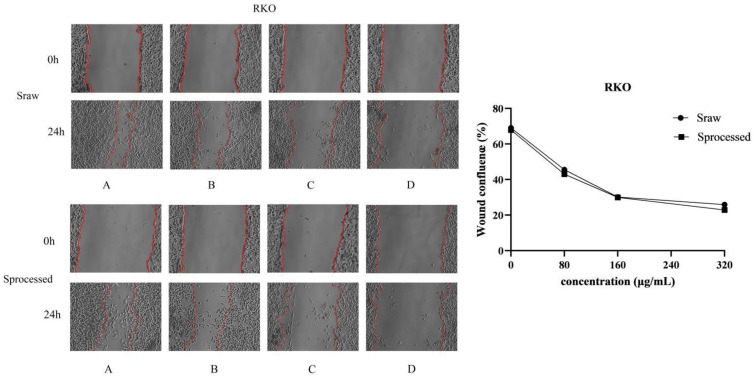
Wound healing assays in RKO cells showing inhibition by *Sraw* and *Sprocessed* (scratch analysis). (**A**): 0 μg/mL; (**B**): 80 μg/mL; (**C**): 160 μg/mL; (**D**): 320 μg/mL.

**Table 1 molecules-27-09046-t001:** Fragment ions of the common components in *Sanguisorba officinalis* L. before and after processing (UHPLC-ESI^-^-MS/MS).

Peak No	t_R_ (min)	Molecular Weight	[M-H]^−^ or [M+HCOOH-H]^-^	MS/MS *m*/*z*	Formula	Compound
2 (N2)	5.699	483.1	484.6	451.3; 338.1; 302.9; 264.1; 220.1	C_30_H_44_O_5_	3,11-dioxo-19α-hydroxylurs-12-en-28-oic acid
5 (N5)	17.852	696.5	695.5	648.8; 487	C_36_H_58_O_10_	2α,3β,19α-trihydroxyurs-12-en-28-acid-β-D-glucopyranosyl ester or isomer
6 (N6)	19.726	766.9	765.6	616.3; 603.5; 585.3; 471.3; 453.1	C_41_H_66_O_13_	3-β[(α-L-arabinopyranosyl) oxy]-29-hydroxy olean-12-en-28-oic acid β-D-glucopyranosyl este
7 * (N7)	20.162	766.4	765.6; 811.3	764.9; 633.8; 616.3; 603.5; 585.3; 207.1; 471.3; 453.3	C_41_H_66_O_13_	Ziyuglycoside Ⅰ
9 * (N9)	25.852	748.4	793.4	657.1; 616.3; 584.6; 585.4; 453.1; 208.1; 190.1	C_41_H_64_O_12_	3-β-O-α-L-arabinosylurs-12,18(19)-dien-28-acid-β-D-glucose ester
10 * (N10)	26.172	748.4	793.4	616.3; 585.3; 471.3; 453.1; 425.8; 377.3; 190.1	C_41_H_64_O_12_	3-β-O-α-L-arabinosylurs-12,19(29)-dien-28-acid-β-D-glucose ester
11 (N11)	26.407	748.4	793.4	656.9; 616.3; 585.3; 541.3; 206.9	C_42_H_68_O_11_	3β-O-α-L-arabinopyranosylurs-12,19-dien-28-β-D-glucopyranosyl ester
12 * (N12)	26.744	634.4	679.4	633.0; 541.3; 471.2; 453.1; 130.7	C_36_H_58_O_9_	3β, l9α-dihydroxyursin-12-en-28-acid-β-D-glucose ester
13 * (N13)	29.599	504.3	503.4	485.5; 467.3; 443.2; 420.7; 313.0; 264.1; 220.1	C_30_H_48_O_6_	1β-hydroxyrosic acid
14 (N14)	32.741	486.7	485.3	423.3; 407.4; 389.0; 373.3; 358.5; 264.1; 220.1; 137.8	C_30_H_46_O_5_	2α,19α-dihydroxy-3-oxo-12-ursen-28-oic acid
15 * (N15)	35.229	488.3	487.3	469.5; 451.3; 406.9; 264.1; 220.1; 206.1	C_30_H_48_O_5_	Euscaphic acid
16 (N16)	35.745	488.7	487.4	469.2; 451.3; 425.3; 264.1	C_30_H_48_O_5_	2α,3α,19α-trihydroxyurs-12-en-28-oic acid
18 * (N18)	37.955	604.5	603.5	585.3; 541.3; 471.3; 453.1; 130.7	C_35_H_56_O_8_	Ziyuglycoside II
19 (N19)	47.551	470.7	469.4	406,9; 264.1; 220.1; 206.1; 130.8; 108.2	C_31_H_50_O_3_	ursolic acid methyl ester
20 (N20)	51.040	484.5	483.3	443.0; 421.3; 355.5; 281.3; 253.4; 147.1; 133.5	C_30_H_44_O_5_	2α,19α-dihydroxy-3-oxo-urs-11,13(18)-dien-28-oic acid

* compared with reference.

**Table 2 molecules-27-09046-t002:** Fragment ions of the common components in *Sanguisorba officinalis* L. before and after processing (UHPLC-ESI^+^-MS/MS).

Peak No	t_R_ (min)	Molecular Weight	[M+H] ^+^	MS/MS *m*/*z*	Formula	Compound
1 (P1)	11.197	812.9	814.5	635.6; 604.6; 545.3; 472.4; 454.5; 428.2	C_42_H_68_O_15_	3-O-β-D-glucopyranosyl-2α,19α-dihydroxyurs-12-en-28-oic acid β-D-glucopyranosyl ester
2 (P2)	13.452	470.3	471.3	453.2; 425.4; 410.9; 340.2; 379.3; 319.8; 238.1; 150.8; 136.6	C_30_H_46_O_4_	Pomeranic acid
4 (P4)	21.420	488.7	489.5	330.0; 314.9; 200.9; 145.1	C_30_H_48_O_5_	1β,2α,3α,19α-tetrahydroxyurs-12-en-28-oic acid
7 (P7)	26.288	748.9	749.9	618.5; 560.5; 473.3; 454.5; 206.9; 130.1	C_42_H_68_O_11_	3β-O-α-L-Arabinopyranosylusr-12,18-dien-28-β-D-glucopyranosyl ester
8 (P8)	26.435	748.9	749.9	618.6; 473.7; 454.6; 206.9; 189.1	C_42_H_68_O_11_	3β-O-α-L-arabinopyranosylusr-12,19(29)-dien-28-β-D-glucopyranosyl ester
9 (P9)	27.032	454.5	455.3	437.4; 409.2; 201.8; 186.8; 179.1	C_30_H_46_O_3_	3β-hydroxyurs-11,13(18)-dien-28-oic acid
10 (P10)	30.019	474.6	475.3	391.2; 286.9; 247.1; 230.7; 204.8; 191.2	C_29_H_46_O_5_	2α,3α,19α-trihydroxyurs-12-en-28-oic-acid or isomer
11 (P11)	36.340	484.3	485.3	467.2; 439.5; 421.1; 403.2; 367.1; 331.2; 282.3; 251.0; 135.2	C_30_H_44_O_5_	2α,19α-dihydroxy-3-oxo-urs-11,13(18)-dien-28-oic acid

**Table 3 molecules-27-09046-t003:** Fragment ions of disappeared components in *Sanguisorba officinalis* L. after processing.

Peak No	t_R_ (min)	[M-H]^−^ or [M+HCOOH-H]^-^	[M+H] ^+^	MS/MS *m*/*z*	Formula	Compound
[1] (N1)	6.235	633.1	/	603.5; 471.3; 453.1; 300.9; 274.7; 248.3	C_36_H_58_O_9_	3β, l9α-dihydroxyursin-12-en-28-acid β-D-glucose ester
[3] (N3)	11.231	695.4	/	649.4; 487.3; 471.3; 453.1; 425.3	C_36_H_58_O_10_	2α,3α,19α-trihydroxyurs-12-en-28-acid-β-D-glucopyranosyl ester or isomer
[3] (P3)	8.782	/	475.3	437.2; 409.3; 391.3; 201.8	C_29_H_46_O_5_	2α,3α,19α-trihydroxyurs-12-en-28-oic-acid

**Table 4 molecules-27-09046-t004:** Fragment ions of new components in *Sanguisorba officinalis* L. after processing.

Peak No	t_R_ (min)	[M-H]^−^ or [M+HCOOH-H]^-^	[M+H] ^+^	MS/MS *m*/*z*	Formula	Compound
[11] (N11)	16.670	695.3	/	649.4; 558.5; 487.3; 425.2; 303.0	C_36_H_58_O_10_	2α,3,19-trihydroxyurs-12-en-28-acid-β-D-glucopyranosyl ester or isomer
[12] (N12)	17.461	695.4	/	649.3; 648.9; 487.3; 475.1	C_36_H_58_O_10_	2α,3,19-trihydroxyurs-12-en-28-acid-β-D-glucopyranosyl ester or isomer
[13] (N13)	34.084	483.2	/	465.1; 421.1; 390.9; 224.9	C_30_H_44_O_5_	3,11-dioxo-19α-hydroxy-urs-12-en-28-oic acid
[15] (N15)	39.216	585.3	/	541.3; 471.3; 453.1; 348.7	C_35_H_54_O_7_	3β-O-α-L-arabinopyranosylusr-12,18-dien-28-acid
[26] (N26)	51.876	451.3	/	404.9; 363.6; 264.8; 252.3	C_29_H_40_O_4_	2α,3β-dihydroxy-28-norurs-12,17,19(20),21-tetraen-23-oic acid
[4] (P4)	19.403	/	458.3	436.8; 391.3; 373.3; 327.2; 276.5; 228.1; 214.9; 179.3; 153.0	C_30_H_48_O_3_	Ursolic acid
[7] (P7)	33.765	/	485.3	439.5; 367.3; 340.9; 336.9; 329.1; 226.2; 201.2; 187.1; 174.6; 149.1; 109.0	C_30_H_44_O_5_	3,11-dioxo-19α-hydroxy-urs-12-en-28-acid
[19] (P19)	52.627	/	455.3	455.1; 440.2; 437.0; 305.0; 283.4; 267.4; 249.4; 201.3; 161.2; 147.2; 119.0; 107.1	C_30_H_46_O_3_	3-oxo-12-en-28-ursolic acid
[20] (P20)	53.923	/	455.3	455.1; 440.2; 391; 283.4; 267.4; 249.4; 201.3; 161.2; 147.2; 119.0; 107.1	C_30_H_46_O_3_	3-oxo-12-en-28-ursolic acid or isomer

## Data Availability

Not applicable.

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
