# Peer review of "Variation of Saponins in Sanguisorba officinalis L. before and after Processing (Paozhi) and Its Effects on Colon Cancer Cells In Vitro"

_molecules, 2022, doi:10.3390/molecules27249046_

Round 1

Reviewer 1 Report

Line 12: by must be replaced by over

Line 18: assay must be replaced by assays

Line 19:  in vitro colon must be replaced by in vitro on

Keywords must be not included in the manuscript title

Line 56:  significance must be replaced by significant

Line 83: MS what does it mean?  Write the complete word

Line 79:  adjustment must be replaced by adjusting

Line 85:  mode must be replaced by modes

Line 112:  were found must be deleted

Figure. 3 and Figure. 4 are not clear and very small

Line 178:  firstly must be replaced by first

Line 179:  secondly must be replaced by then

Line 181:  changes of content must be replaced by changes in content

Line 299:  difference must be replaced by different

Line 307:  cell must be replaced by cells

Author Response

Response to Reviewer 1 Comments

Point 1: Line 12: by must be replaced by over

Response 1: We deeply appreciate the reviewer’s suggestion. We have studied comments carefully and have made line 12 correction. Line 12, the revised are as follows: “The incidence of colon cancer is increasing year over year, which seriously affects human health and quality of life, in recent years.”

Point 2: Line 18: assay must be replaced by assays

Response 2: Thank you for your comment. And we agreed your points. Line 18, the revised are as follows: “Then, MTT and cell migration assays were used to preliminarily explore the effects of saponins in vitro colon cancer cells.”

Point 3: Line 19:  in vitro colon must be replaced by in vitro on

Response 3: Thank you for your precious comments and advice. We apologize for this deficiency. Line 19, the revised are as follows: “Then, MTT and cell migration assays were used to preliminarily explore the effects of saponins in vitro on colon cancer cells.”

Point 4: Keywords must be not included in the manuscript title

Response 4: We are grateful for the suggestion. We are also extremely grateful to reviewer for pointing out this problem. Therefore, keywords revised are as follows: “Keywords: activity study; potential ingredients; traditional Chinese medicine; qualitative analysis; content determination; UHPLC-MS/MS”

Point 5: Line 56:  significance must be replaced by significant

Response 5: Thank you for your careful reading of our manuscript. We apologize for this deficiency. And we also agreed your points. Therefore, the revised content (Line 73 on revised manuscript) are as follows: “Therefore, it is significant to analysis the changed saponins components before and after processing to clarify its pharmacodynamic material basis.”

Point 6: Line 83: MS what does it mean?  Write the complete word

Response 6: Thank you very much for your precious comment of our manuscript. We sincerely apologize for the inconvenience of reading caused by word problems. MS means Mass Spectrometry. And the complete word has been added on revised manuscript (Line 112) as follows: “Because the complexity of TCM ingredients, to achieve as much Mass Spectrometry (MS) information as possible, the compounds were investigated under positive and negative ion modes.”

Point 7: Line 79:  adjustment must be replaced by adjusting

Response 7: We are grateful for the suggestion. We agree with the comment and re-wrote the words in the revised manuscript (Line 108) as the following: “the chromatographic conditions were optimized by using different compositions of mobile phase and adjusting of gradient elution.”

Point 8: Line 85:  mode must be replaced by modes

Response 8: Thank you for your suggestion. We have modified this expression in the revised manuscript of corresponding position. And our revised content (Line 114 on revised manuscript) are as follows: “the compounds were investigated under positive and negative ion modes.”

Point 9: Line 112:  were found must be deleted

Response 9: Thank you for underlining this deficiency. “were found” was deleted according to the information suggested by the reviewer.

Point 10: Figure. 3 and Figure. 4 are not clear and very small

Response 10: Thank you for your precious comments and advice. Those comments are all valuable and very helpful for revising and improving our paper. We apologize for the problems with picture clarity in the original manuscript. In addition, according to the reviewer’s comment, Figure. 3 and Figure. 4 were improved. We agree with the comment and re-provided the figures in the revised manuscript as the following:

Figure. 3 Mass spectrum and structure of disappeared components in Sanguisorba officinalis L. after processing 

A: 3β, l9α-dihydroxyursin-12-en-28-acid-β-D-glucose ester; B: 2α,3α,19α-trihydroxyurs-12-en-28-acid-β-D-glucopyranosyl ester or isomer; C: 2α,3α,19α-trihydroxyurs-12-en-28-oic-acid

Figure.4 Mass spectrum and structure of new components in Sanguisorba officinalis L. after processing

A: 2α,3,19-Trihydroxyurs-12-en-28-acid-β-D-glucopyranosyl ester or isomer; B: 2α,3,19-Trihydroxyurs-12-en-28-acid-β-D-glucopyranosyl ester or isomer; C: 3,11-dioxo-19α-hydroxy-urs-12-en-28-oic acid; D: 3β-O-α-L-arabinopyranosylusr-12,18-dien-28-acid; E: 2α,3β-dihydroxy-28-norurs-12,17,19(20),21-tetraen-23-oic acid; F: ursolic acid; G: 3,11-dioxo-19α-hydroxy-urs-12-en-28-acid; H: 3-oxo-12-en-28-ursolic acid or isomer; I: 3-oxo-12-en-28-ursolic acid or isomer

Point 11: Line 178:  firstly must be replaced by first

Response 11: Thank you for your suggestion. As suggested by reviewer, we have replaced the suggested content in the revised manuscript (Line 207). The revised content are as follows: ”Based on the above principles, during the processing, the disubstituted saponins may first be converted into the monosubstituted components, and then, into the sapogenin components.”

Point 12: Line 179:  secondly must be replaced by then

Response 12: Thank you for your comment, and the revised content (Line 208 on revised manuscript) are as follows: ”Based on the above principles, during the processing, the disubstituted saponins may first be converted into the monosubstituted components, and then, into the sapogenin components.”

Point 13: Line 181:  changes of content must be replaced by changes in content

Response 13: We deeply appreciate the reviewer’s suggestion. According to the reviewer’s comment, we have modified this content in the revised manuscript of corresponding position. The revised (Line 210 on revised manuscript) content are as follows: “Then it may cause the changes in content and pharmacological effects, and differences in component content of Sraw and Sprocessed will also affect the efficacy.”

Point 14: Line 299:  difference must be replaced by different

Response 14: We are grateful for the suggestion. We have modified this expression in the revised manuscript of corresponding position. And our revised content (Line 328 on revised manuscript) are as follows: “From the perspective of the overall compound changes, 17 different compounds were screened through UHPLC-MS/MS combined with multivariate statistical analysis.”

Point 15: Line 307:  cell must be replaced by cells

Response 15: Thank you for your precious comments and advice. We apologize for the word problems in the original manuscript. We have made corrections in revised manuscript. And our revised content (Line 336 on revised manuscript) are as follows: “Therefore, in our study, we combined cells in vitro provide novel insight for S-Saponins before and after Paozhi.”

Reviewer 2 Report

This article is about Changes of saponins in Sanguisorba officinalis L. before and after processing (Paozhiand its effects on colon cancer cells in vitro. Impressive Results part. Please see my suggestions:

L35. Why is urgent? All diseases must be cured as soon as possible, but natural medicine has notal the the perfect medicine/remedy. Since ancient times https://www.researchgate.net/publication/286442576_Between_Religion_and_Science_Some_Aspects_Concerning_Illness_and_Healing_in_Antiquity, the nature was used for treatment of different disease, but none of the plant-based compounds are not healing cancer of any type.

Introduction must be better developed. The authors have detailed the saponins, but about colon cancer (CC) very few data in L31-34. Regarding the CC, it must be detailed as follows:

- What importance heave early diagnosis of CC?

- How can it be diagnosed earlier (techniques, apparatus, approaches)? 

- Which treatments are provided in protocols (i.e. immunohistochemical and histoenzymatic techniques, etc.)? A separate paragraph must be dedicated to CC diagnostics and treatment options. In this regard, I suggest checking and referring to: https://pubmed.ncbi.nlm.nih.gov/26662146/

L63-74. As the topic is not a new one, in the last paragraph of Introduction, please highlight better the special aspects that your study brings to the field, and what differentiate this paper from other in the same topic.

After L322, please add the strengths and the weakness of your Results/research.

Conclusions section. L 408 please remove “ In conclusion, …”, as it is obvious.

Author Response

Response to Reviewer 2 Comments

Point 1: This article is about Changes of saponins in Sanguisorba officinalis L. before and after processing (Paozhi) 2 and its effects on colon cancer cells in vitro. Impressive Results part. Please see my suggestions:

L35. Why is urgent? All diseases must be cured as soon as possible, but natural medicine has notal the the perfect medicine/remedy. Since ancient times https://www.researchgate.net/publication/286442576_Between_Religion_and_Science_Some_Aspects_Concerning_Illness_and_Healing_in_Antiquity, the nature was used for treatment of different disease, but none of the plant-based compounds are not healing cancer of any type.

Response 1: Thank you for your careful review. The reviewer gives an accurate summary of our work and brings forward constructive questions. To be more clearly and in accordance with the reviewer concerns, we have added a more detailed interpretation regarding “Why is urgent?”.We include the following description to the introduction with highlight on Line 43-52 in the revised manuscript, meanwhile, we have added corresponding new references with highlight on Line 491-497 in the revised manuscript:

“At present, the treatment of colon cancer mainly focuses on surgery, radiation therapy, and chemotherapy. However, the serious side effects of chemotherapy drugs and other disadvantages limit its clinical application [5]. Traditional Chinese medicine (TCM) has shown anti-tumor potential through “multi-component, multi-target, and multi-pathway” compared with chemotherapy drugs, which can improve the disease process through a variety of ways [6]. Furthermore, researchers have been attracted by TCM’s low cost, stable curative effect, and minimal side effects, leading them to explore TCM’s potential and application prospects for treating tumors [7]. Therefore, it is urgent to find natural medicine to meet the treatment needs of the complex pathological mechanism of colon cancer and lower the risk of colon cancer.”

References

  1. Gelibter, A.J.; Caponnetto, S.; Urbano, F.; Emiliani, A.; Scagnoli, S.; Sirgiovanni, G.; Napoli, V.M.; Cortesi, E. Adjuvant Chemotherapy in Resected Colon Cancer: When, How and How Long? Surg. Oncol. 2019, 30, 100–107, doi: 10.1016/j.suronc.2019.06.003.
  2. Luo, H.; Vong, C. T.; Chen, H.; Gao, Y.; Lyu, P.; Qiu, L.; Zhao, M.; Liu, Q.; Cheng, Z.; Zou, J.; Yao, P.; Gao, C.; Wei, J.; Ung, C. O. L.; Wang, S.; Zhong, Z.; Wang, Y. Naturally occurring anti-cancer compounds: shining from Chinese herbal medicine. Chinese medicine. 2019, 14, 48. doi: 10.1186/s13020-019-0270-9. 
  3. Liu, Y.;Yang, S.; Wang, K.; Lu, J.; Bao, X.; Wang, R.; Qiu, Y.; Wang, T.; Yu, H. Cellular senescence and cancer: Focusing on traditional Chinese medicine and natural products. Cell proliferation. 2020, 53(10), e12894. doi: 10.1111/cpr.12894.

Point 2: Introduction must be better developed. The authors have detailed the saponins, but about colon cancer (CC) very few data in L31-34. Regarding the CC, it must be detailed as follows:

- What importance heave early diagnosis of CC?

- How can it be diagnosed earlier (techniques, apparatus, approaches)? 

- Which treatments are provided in protocols (i.e. immunohistochemical and histoenzymatic techniques, etc.)? A separate paragraph must be dedicated to CC diagnostics and treatment options. In this regard, I suggest checking and referring to: https://pubmed.ncbi.nlm.nih.gov/26662146/

Response 2: Thank you for your suggestion. We have carefully addressed all the reviewer's concerns. And according to your suggestion the detailed about colon cancer has been replenished. We include the following description to the introduction with highlight on Line 32-42 in the revised manuscript, meanwhile, we have added corresponding new references with highlight on Line 485-490 in the revised manuscript:

“In recent years, colon cancer has become a common malignant tumor of the digestive tract, which seriously threatens human health [1]. Early diagnosis of colon cancer can effectively reduce the incidence and mortality of colon cancer, which will mean that more patients with colon cancer will have the opportunity to have a better chance of long-term survival due to early detection, early diagnosis and early treatment. In addition, it has been reported that the five-year survival rate of patients with colon cancer is more than 90 % if it is prevented and treated in an early stage [2]. Currently, there are several kinds of early diagnosis ways of colon cancer including fecal immunochemical test (FIT), colonoscopy, guaiac-based fecal occult blood test, multitargeted stool DNA test (FIT-DNA), flexible sigmoidoscopy, CT colonography, etc [3]. In addition, Annamaria pallag et al. used immunohistochemical and histoenzymatic techniques to monitor colon cancer cells for achieving the purpose of diagnoses [4].”

References

  1. Otani, K.;Kawai, K.;Hata, K.; Tanaka, T.; Nishikawa, T.; Sasaki, K.; Kaneko, M.; Murono, K.; Emoto, S.; Nozawa, H. Colon cancer with perforation. Surgery today. 2019, 49(1), 15–20. doi: 10.1007/s00595-018-1661-8.
  2. Pawlik T. M. Colon Cancer. Surgical oncology clinics of North America. 2018, 27(2), xiii–xiv. doi: 10.1016/j.soc.2017.11.013.
  3. Pallag, A.; RoÅŸca, E.; Å£iÅ¢, D. M.; MuÅ¢iu, G.; Bungău, S. G.; Pop, O. L. Monitoring the effects of treatment in colon cancer cells using immunohistochemical and histoenzymatic techniques.Romanian journal of morphology and embryology = Revue roumaine de morphologie et embryologie. 2015, 56(3), 1103–1109.

Point 3: L63-74. As the topic is not a new one, in the last paragraph of Introduction, please highlight better the special aspects that your study brings to the field, and what differentiate this paper from other in the same topic.

Response 3: We are grateful to the reviewer for this comment and realize that these differences may not have been expressed clearly enough in the previous manuscript. We have made improvements to the original manuscript in order to make clearer the special aspects that our study brings to the field. We include the following description to the introduction with highlight on Line 75-78, Line 80-84 and Line 87-98 in the revised manuscript:

“In recent years, with the development of technology and the change of processing technology, the clinical application of Sanguisorba officinalis L. saponins proccessed (Sprocessed) products has also been deepened [18, 19]. The ancient processing methods of S. officinalis include roasting, vinegar, frying, simmering, wine, charcoal, etc [20]. Among of them, raw products and charcoal products have been used to this day [21]. The chemical composition and trace elements contained in S. officinalis will change after processing. Traditionally, it is believed that S. officinalis will enhance its corresponding efficacy after processing [22]. Because TCM has a very complex compounds, and the chemical composition may change after processing. Therefore, it is significant to analysis the changed saponins components before and after processing to clarify its pharmacodynamic material basis. At present, ultra-high-performance liquid chromatography-tandem mass spectrometry (UHPLC-MS/MS) technology has been used to study the active ingredients of complex systems of natural medicines, given its high sensitivity and resolution, good selectivity, and short analysis time [23, 24]. However, the studies on the composition changes of S. officinalis before and after processing mainly focus on a certain component [25, 26], there are few reports of studying multiple component changes simultaneously. Obviously, the pharmacological activity and intrinsic quality of natural medicine cannot be fully described by considering only one compounds. Therefore, it is necessary to monitor as many bioactive components of S. officinalis under Paozhi as possible to ensure its quality and efficacy. It can be more fully reflected the differences of pharmacodynamic substances before and after processing by accurately controlling the changes of saponin components and the screening of different compounds.

To the best of our knowledge, no study has comprehensively analyzed the different compounds in raw and processed S. officinalis. In our study, for the first time, the differential components within differently processed S. officinalis saponins products were analyzed comprehensively and the possible reasons for the differences in efficacy were considered. Meanwhile, this study has also some innovative content in the application of mass spectrometry, the instrumentation of UHPLC-MS/MS was used to rapidly detect and identify the components in S. officinalis, and at the same time multivariate statistical analysis approach was established to discover the changes in chemical composition after processing. Consequently, a convenient and systematic way was established to investigate the changes in different compounds of Sraw and Sprocessed. Meanwhile, this study could serve as a theoretical basis for intensive mechanistic studies of S. officinalis processing and reasonable clinical applications. Moreover, we preliminarily evaluated the effects of Sraw (Sanguisorba officinalis L. saponins raw) and Sprocessed (Sanguisorba officinalis L. saponins processed) on the growth of colon cancer cell lines. We tried from Paozhi perspective to reveal the changed compounds and possible transformation pathways in S. officinalis, which may be the core link to help explain its for colon cancer prevention. Furthermore, it also might promote the development of effective disease modifying TCM extracts.”

Point 4: After L322, please add the strengths and the weakness of your Results/research.

Response 4: We are grateful for the suggestion. We are also extremely grateful to reviewer for pointing out this problem. According to your advice, we have now added the strengths and the weakness that make the statement of the article clearer. We have modified this content in the revised manuscript of corresponding position (Line 351-357). And our revised content are as follows:

“In this study, an accurate, reliable and simple method was established for the analysis of the saponins of S. officinalis before and after processing to illuminate its composition difference and content change of saponins. Moreover, difference of Sraw and Sprocessed against colon cancer were verified by simple in vitro cell experiments, which provided certain guiding significance for the discovery of different compounds of S. officinalis. Meanwhile, the results of our study would be a helpful reference for further explaining the mechanisms underlying the effect of processing process on S. officinalis and help us to find the potential bioactive components for treating colon cancer. In addition, our study suggested that UHPLC-MS/MS method was useful in studies of active compounds in TCM extracts. However, its anti-tumor mechanism still needs to be further studied. In the next step, we will carry out the research of these different compounds to deeply find out the in vivo process and the mechanisms for treating colon diseases.”

Point 5: Conclusions section. L 408 please remove “In conclusion, …”, as it is obvious.

Response 5: Thank you for underlining this deficiency. Our deepest gratitude goes to you for your careful work and thoughtful suggestions that have helped improve this paper substantially. This section was revised and modified according to the information suggested by the reviewer. We include the following description with highlight on Line 444-447 of the revised manuscript: “This study demonstrated that a qualitative and quantitative analysis methods was successfully developed to screen the content change and iconic compounds of S. officinalis before and after Paozhi through UHPLC-ESI/MS combined with multivariate statistical analysis technology. “
